# Identification of GC-rich LAT genes in birds

**Sarka Janusova**[1,2☯], **Veronika Krchlikova**[3☯], **Tomas Hron**[3], **Daniel Elleder**[3]*, **Ondrej Stepanek**[1]*

**1** Laboratory of Adaptive Immunity, Institute of Molecular Genetics of the Czech Academy of Sciences, Prague, Czech Republic, **2** Faculty of Science, Department of Cell Biology, Charles University, Prague, Czech Republic, **3** Laboratory of Viral and Cellular Genetics, Institute of Molecular Genetics of the Czech Academy of Sciences, Prague, Czech Republic

☯ These authors contributed equally to this work.
* daniel.elleder@img.cas.cz (DE); ondrej.stepanek@img.cas.cz (OS)

**Data Availability Statement:** Gene sequences were deposited in the DDBJ/ENA/GenBank databases under the accession numbers TPA: BK061375-BK061384. Other relevant data are within the paper and its Supporting Information files.

## Abstract

Linker for activation of T cells (LAT) plays a key role in T-cell antigenic signaling in mammals. Accordingly, LAT orthologues were identified in the majority of vertebrates. However, *LAT* orthologues were not identified in most birds. In this study, we show that *LAT* gene is present in genomes of multiple extant birds. It was not properly assembled previously because of its GC-rich content. LAT expression is enriched in lymphoid organs in chicken. The analysis of the coding sequences revealed a strong conservation of key signaling motifs in LAT between chicken and human. Overall, our data indicate that mammalian and avian *LAT* genes are functional homologues with a common role in T-cell signaling.

## Introduction

Linker for activation of T-cells (LAT) is a key transmembrane adaptor protein in the T-cell antigen receptor (TCR) signaling pathway [1–4]. It consists of a very short extracellular domain, a transmembrane domain with two palmitoylated cysteine residues, and an intracellular tail containing several signaling phosphotyrosine motifs [1, 2, 5, 6].

The importance of LAT was first demonstrated in LAT-deficient Jurkat cell lines, JCaM2 and ANJ3. These cell lines have impaired calcium signaling and ERK phosphorylation after TCR activation [2, 7]. LAT-deficient mice show a severe block in early thymic T-cell development, resulting in a low number of peripheral T cells [4, 8, 9] and defective distal TCR signaling [2, 4, 10].

Four of the nine conserved tyrosine residues (Y132, Y171, Y191, and Y226, numbering according to human LAT throughout the manuscript) in the intracellular part of LAT were identified as important docking sites for several downstream molecules in the TCR signaling cascade, such as Grb2, Gad, and PLCγ1 [1–3, 6, 10–13]. Phosphorylation of these tyrosine residues via ZAP-70 kinase is a crucial step for triggering downstream signaling pathways [1, 2, 13].

Phosphorylated Y132 is the only motif in LAT that recruits PLCγ1. Accordingly, Y132 was shown to be essential for TCR-downstream signaling in Jurkat cells as well as in mice [2, 9, 11–14]. Surprisingly, Y132 is a suboptimal substrate for ZAP-70 due to the glycine residue in position 131 in all known LAT sequences in tetrapods [12, 15]. It has been proposed that inefficient

**Funding:** This project was supported by the National Institute of virology and bacteriology (Programme EXCELES, ID Project No. LX22NPO5103) - Funded by the European Union - Next Generation EU and Charles University Grant Agency (Project No. 984120 to SJ). The funders had no role in study design, data collection and analysis, decision to publish, or preparation of the manuscript.

**Competing interests:** The authors have declared that no competing interests exist.

phosphorylation of Y132 by ZAP-70 introduces a delay, which is important for ligand discrimination via a kinetic proofreading mechanism [16]. Some fish species have aspartate or glutamate residue at the position 131, which makes Y132 an optimal substrate for ZAP-70 [12]. The biological significance of the interspecies differences in the amino acid preceding Y132 is unclear, but it has been proposed that some fish may need a negatively charged amino acid at the 131 position to compensate for low kinase activity in a cold environment [12].

LAT deficiency in humans manifests with an early onset of severe combined immunodeficiency (SCID) with recurrent infections [4, 17]. Patients have impaired T-cell function with a skewed response to Th2, reduced number of B cells, and increased number of γδ T cells. These conditions are lethal in the early age of patients unless bone marrow transplantation is performed [17].

LAT was described in tetrapods, fish and in two palaeognath birds (*Dromaius novaehollandiae* and *Apteryx rowi*) [12]. Surprisingly, in the avian clade Neognathae, which contains the vast majority of extant bird species, *LAT* orthologue has not been identified so far [12]. This opened the possibility that T-cell signaling does not rely on this adaptor protein in most birds.

We used our previous experience in the search for avian genes that are difficult to sequence and identify due to their high GC content [18]. In this study, we show that LAT is in fact encoded in the genome of neognath birds including chicken *(Gallus gallus)*. Moreover, avian LAT contains conserved signaling motifs and is prevalently expressed in lymphoid tissues. This strongly suggests that LAT plays the same role in birds as in mammals.

## Material and methods

### Identification and phylogenetic analysis of avian LAT orthologues

For the identification and assembly of avian LAT sequences, we used 'raw' next generation sequencing (NGS) data available in the Sequence Read Archive (SRA) of the National Center for Biotechnology information (NCBI). Both RNA-seq and genomic SRA datasets were utilized. Sequencing reads originating from avian LAT were identified by BLASTN searches using various known avian and non-avian LAT sequences as baits. The collected reads of each avian species were consequently assembled using DNASTAR Lasergene SeqMan Pro and CLC genomics workbench software. Nucleotide sequence data reported are available in the Third Party Annotation Section of the DDBJ/ENA/GenBank databases under the accession numbers TPA: BK061375-BK061384 (BK061375 –*Apteryx rowi*, BK061376 –*Struthio camelus*, BK061377 –*Gallus gallus*, BK061378 –*Numida meleagris*, BK061379 –*Meleagris gallopavo*, BK061380 –*Anas platyrhynchos*, BK061381 –*Anser cygnoides*, BK061382 –*Taeniopygia guttata*, BK061383 –*Parus major*, BK061384 –*Pipra filicauda*).

Phylogenetic analysis was conducted for all newly identified avian LAT sequences together with following known vertebrate orthologues: *Homo sapiens* (NCBI RefSeq ID: NP_001014987.1), *Pan troglodytes* (XP_009439105.1), *Mus musculus* (NP_034819.1), *Equus caballus* (XP_023471840.1), *Ornithorhynchus anatinus* (XP_028913760.1), *Anolis carolinensis* (XP_008115721.1), *Alligator mississippiensis* (XP_014453735.1), *Xenopus tropicalis* (XP_031749589.1), *Danio rerio* (NP_001137156.1), *Dromaius novaehollandiae* (XP_025978835.1). Sequence alignment was performed using Mafft v7.487 with default settings. More specifically, amino acid sequences were aligned separately for i) mammals, ii) reptiles, iii) fish and amphibia, and iv) birds. The resulting alignments were merged into a single alignment using MAFFT L-INS-i algorithm. Phylogenetic tree was reconstructed using PhyML with 500 bootstrap replicates, LG substitution model, site rate variation under gamma distribution with 4 categories, and Subtree-Pruning-Regrafting (SPR) searching operations.

## Polymerase chain reaction (PCR)

Total RNA was isolated using TRI reagent (Sigma-Aldrich) from spleen tissue of Brown Leg-horn chicken (*Gallus gallus*) within a previous project [19]. The reverse transcription was per-formed using the SMART RACE (Clontech) procedure. Chicken LAT (cLAT) amplification was performed using a mixture of two polymerases (1:200 Deep Vent: Taq; both from NEB) and primers 5'–TCCCAAAGGCGGCGGT and 5'–CGTCTTCTCAGGTTGCGTCAGC. 5 μl of cDNA were used in a 20 μl reaction. The PCR program was set to 95˚C for 2 min, followed by 30 cycles of 95˚C for 30 s and 65˚C for 10 min. The long elongation times allow for efficient amplification of GC-rich sequences. PCR products were separated on 1% agarose gel. Band of interest was purified using QIAEX II Gel Extraction Kit (Qiagen) and submitted for Sanger sequencing (SEQme).

## Relative mRNA expression in different organs

For *in silico* determination of relative expression of cLAT, we employed a series of tissue-spe-cific RNA sequencing datasets, publicly available in SRA under the study ID PRJEB12891. cLAT reads were extracted from individual datasets using BLASTN search with default param-eters and bit score threshold 100. Consequently, numbers of extracted reads for each tissue were calculated and expressed as normalized RPKM (Reads per kilobase per million mapped reads) values.

## Ethics statement

There are no relevant ethical concerns in this study. No human subjects were studied and no experiments on animals were performed within this project.

## Results

To identify *LAT* genes in avian genomes, we used similar approach as in our previous searches for avian GC-rich 'hidden' genes [18–20]. We performed homology-based searches of the SRA datasets in NCBI (mostly represented by Illumina RNA-seq data) to assemble full coding sequences of LAT in ten bird species including chicken. To further verify that these represent true *LAT* orthologues, we inferred a phylogenetic relationship with selected vertebrate *LAT* sequences. The clustering of mammalian, avian, and reptilian *LAT* sequences followed known evolutionary relationships (Fig 1).

All newly identified avian *LAT* sequences have high GC content and are richer in G/C stretches than their non-avian orthologues (Fig 2). These characteristics are the main cause of their resistance to PCR amplification [18]. To overcome these difficulties, we used our previ-ously optimized PCR conditions (see Methods) to amplify the full-length chicken LAT (cLAT) coding sequence from splenic RNA, which was further confirmed by sequencing.

As a further line of evidence pointing to direct orthology between avian and mammalian *LAT* genes, we could identify synthetic gene order in several species where larger genomic con-tigs were available. Most clearly annotated example is kiwi (*A. rowi*) genomic contig NW_020447579, where *LAT* lies next to SPNS1, a gene whose orthologue is also genomic neighbor of human *LAT*.

The alignment of the amino acid sequence cLAT with human LAT (hLAT), and zebrafish (*Danio rerio*) LAT (zLAT) revealed that the conserved palmitoylation cysteines, phosphoryla-tion tyrosine motifs [21] are present in all three sequences (Fig 3). cLAT includes the glycine at the position 131, which is present in hLAT, but not in the zLAT, suggesting that the PLCγ-binding tyrosine in cLAT is not an optimal substrate for ZAP70-mediated phosphorylation

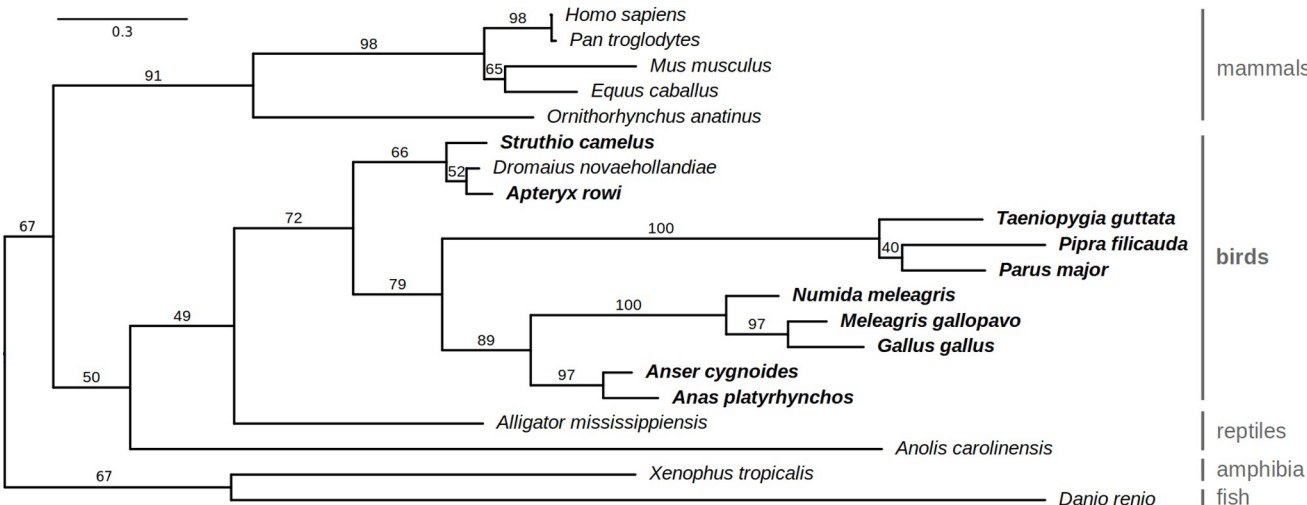

**Fig 1. Phylogenetic relationship of _LAT_ genes from birds and other vertebrates.** The maximum likelihood tree was constructed from the newly identified avian _LAT_ sequences together with selected orthologues from other vertebrates. Bootstrap support values (percent of 500 replicates) are shown above the branches. Species with newly identified LAT sequence are in bold. The scale bar above indicates the number of substitutions per site.

[12]. cLAT as well as zLAT do not contain a proline-rich motif PIPRSP, which mediates the interaction between LAT and the SH3 domain of LCK and facilitates the down-stream TCR signaling in mammals [21]. Despite this missing motif, all three LAT sequences contain numerous prolines in the N-terminal part of their intracellular domains. Overall, similarities

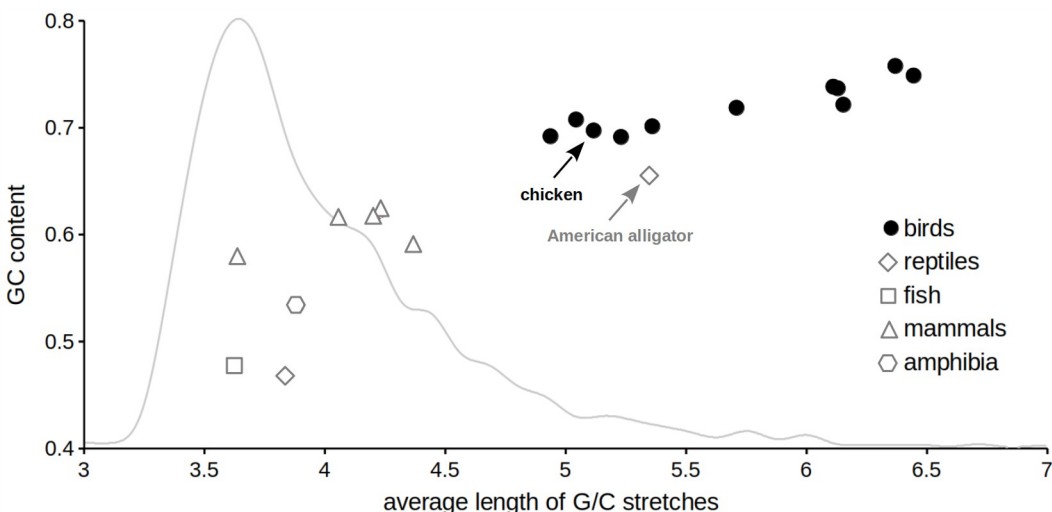

**Fig 2. _LAT_ gene exhibits a high GC content and long G/C stretches in birds.** Comparison of GC content and the presence of G/C stretches in _LAT_ genes from birds and other vertebrates. Dot plot was generated based on the coding sequences of _LAT_ gene from vertebrate species shown in Fig 1. GC content is plotted against average length of sequence stretches containing G or C nucleotides. G/C-stretch was defined as an undisrupted sequence of at least three consecutive G or C nucleotides [18]. To allow comparison with all annotated chicken genes, a histogram showing the distribution of average G/C-stretch length in the chicken RefSeq gene category is depicted by a gray line. Chicken RefSeq genes comprise the complete set of approximately six thousand chicken coding sequences longer than 299 nucleotides. The chicken (_Gallus gallus_) and American alligator (_Alligator mississippiensis_), which is the closest relative to birds from included reptile species, are highlighted by arrows.

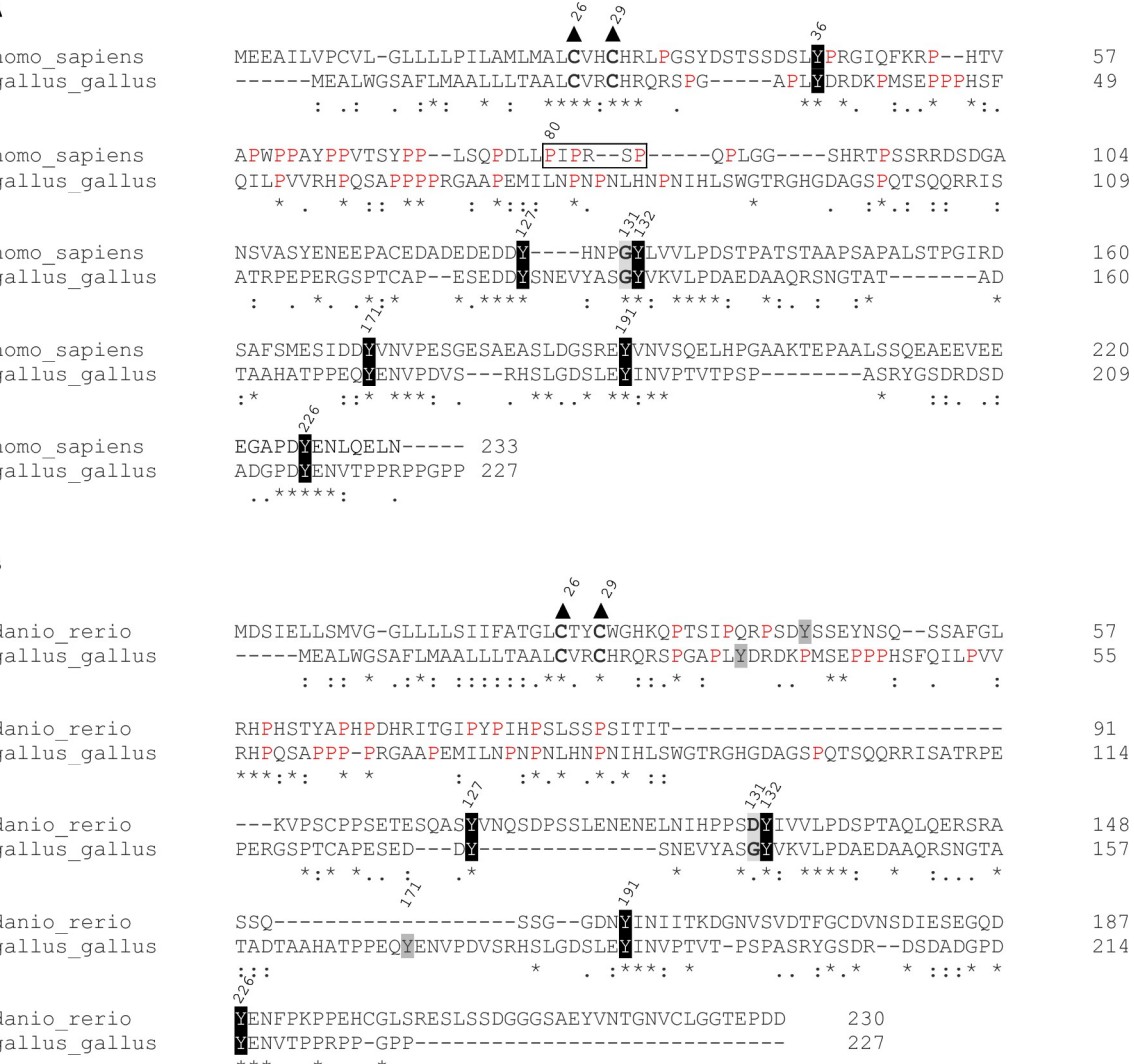

**Fig 3. cLAT contains conserved palmitoylation and signaling motifs.** The alignments of amino acid LAT sequences of human and chicken (A) and zebrafish and chicken (B) are shown. Conserved tyrosine residues are highlighted in black, glycine and/or aspartate at position 131 is highlighted in grey, conserved palmitoylated cysteines are in bold with black triangle, prolines in the proline-rich sequence (amino acids 30–100 of hLAT) are in red, proline-rich LCK-binding motif is highlighted in the hLAT sequence.

in the conserved amino acid residues suggest that cLAT is a functional orthologue of hLAT and zLAT.

LAT is mostly expressed in immunological-relevant organs, such as thymus, tonsils, or spleen in mammals [1, 22]. Using the analysis of publicly available transcriptomic data, we documented a similar expression profile in chicken (Fig 4).

## Discussion

LAT plays an instrumental role in the adaptive immune system of mammals. However, the respective gene was not identified in extant birds with the exception of a relatively small clade of Palaeognathae. In this study, we identified *LAT* orthologues in the genome of several birds from the numerous clades of Neognathae, including chicken. This suggests that *LAT* is generally present in avian species. The most probable reason, why *LAT* was not identified in

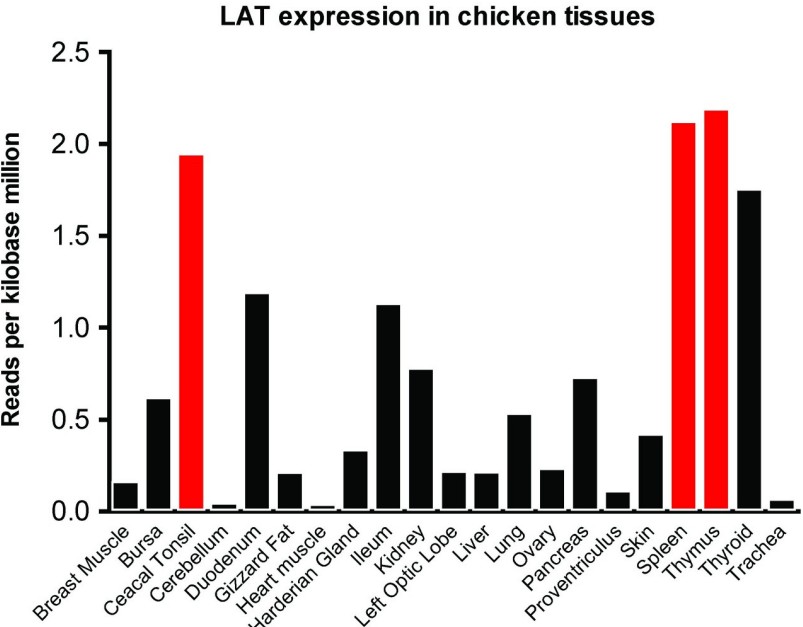

**Fig 4. Expression of cLAT is enriched in immune-related organs.** The expression levels of cLAT mRNA was determined *in silico* from SRA data (see Methods). Immune-related organs are depicted by red bars.

Neognathae birds before, is the high content of GC-rich sequences. These represent the main obstacle in successful amplification and assembly of a subset of avian genes [18, 23–25]. The potential evolutionary cause for this high GC content in avian *LAT* is not clear. Phylogenetic analysis showed that avian *LAT* sequences separate from those present in mammalian, fish or reptilian genomes, as expected from the known evolutionary relationships.

Furthermore, *LAT* expression is enriched in the lymphoid organs in chicken similarly to mammals [26] suggesting a common function in both animal classes. On the protein level, cLAT and hLAT share striking similarities in the functionally relevant parts such as conserved cysteine and tyrosine motifs.

Overall, we conclude that missing LAT in the annotated genomes of neognath birds was caused by technical issues originating from its high GC content. The comparison of the gene expression patterns and the primary structure of the protein products in chicken, human and zebrafish revealed striking similarities. For this reason, we conclude that mammalian, fish and avian *LAT* genes are functional orthologues.

## Author Contributions

**Conceptualization:** Sarka Janusova, Veronika Krchlikova, Tomas Hron, Daniel Elleder, Ondrej Stepanek.

**Formal analysis:** Sarka Janusova, Veronika Krchlikova, Tomas Hron, Daniel Elleder.

**Funding acquisition:** Sarka Janusova, Daniel Elleder, Ondrej Stepanek.

**Investigation:** Sarka Janusova, Veronika Krchlikova, Tomas Hron, Daniel Elleder.

**Methodology:** Veronika Krchlikova, Tomas Hron, Daniel Elleder.

**Project administration:** Daniel Elleder, Ondrej Stepanek.

**Resources:** Veronika Krchlikova, Daniel Elleder.

**Supervision:** Daniel Elleder, Ondrej Stepanek.

**Validation:** Sarka Janusova, Veronika Krchlikova, Tomas Hron, Daniel Elleder.

**Visualization:** Sarka Janusova, Veronika Krchlikova, Tomas Hron.

**Writing – original draft:** Sarka Janusova, Veronika Krchlikova, Tomas Hron, Daniel Elleder, Ondrej Stepanek.

**Writing – review & editing:** Sarka Janusova, Veronika Krchlikova, Tomas Hron, Daniel Elleder, Ondrej Stepanek.

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
