## [Decision Letter · Decision Letter 0]

7 Feb 2023

PONE-D-22-34260Identification of GC-rich LAT genes in birdsPLOS ONE

Dear Dr. Stepanek,

Thank you for submitting your manuscript to PLOS ONE. After careful consideration, we feel that it has merit but does not fully meet PLOS ONE’s publication criteria as it currently stands. Therefore, we invite you to submit a revised version of the manuscript that addresses the points raised during the review process.

1) Address the minor concerns raised by reviewer #1 by making the appropriate changes to the figures/text. 

We look forward to receiving your revised manuscript.

Kind regards,

Sebastian D. Fugmann, Ph.D.

Academic Editor

PLOS ONE

Journal Requirements:

"This project was supported by the National Institute of virology and bacteriology (Programme EXCELES, ID Project No. LX22NPO5103) - Funded by the European Union - Next Generation EU and Charles University Grant Agency (Project No. 984120 to SJ)."

"This project was supported by the National Institute of virology and bacteriology (Programme EXCELES, ID Project No. LX22NPO5103) - Funded by the European Union - Next Generation EU (to OS) and Charles University Grant Agency (Project No. 984120 to SJ)."

Reviewers' comments:

Reviewer's Responses to Questions

**Comments to the Author**

1. Is the manuscript technically sound, and do the data support the conclusions?

Reviewer #1: Yes

2. Has the statistical analysis been performed appropriately and rigorously? 

Reviewer #1: N/A

3. Have the authors made all data underlying the findings in their manuscript fully available?

Reviewer #1: Yes

4. Is the manuscript presented in an intelligible fashion and written in standard English?

Reviewer #1: Yes

5. Review Comments to the Author

Reviewer #1: First, my apologies to the authors. My January happened to be exceptionally busy.

LAT is a critical signaling adapter molecule in the T cell receptor signaling pathway that very likely plays a role in ligand discrimination. LAT orthologues have been identified in various species, including fish. Reports on LAT in bird species have been lacking.

In the submitted manuscript, the authors capitalize on their experience searching for and analyzing genes with high GC content. The authors describe the identification of chicken LAT and postulate – based on sequence homology to human LAT and chicken LAT expression – that chicken LAT full fills a similar function as mammalian in LAT in T cell signaling.

In sum, this is a clearly presented, brief definitive report on the identification of chicken LAT that is informative for the T cell signaling community.

The authors should enhance the manuscript in a couple of manners:

1. The presentation of the amino acid sequence alignment in figure 3 can be improved. It should contain the numbering of e.g. tyrosine residues that are mentioned in the text.

2. The authors should also perform a comparison between chicken and fish LAT (with potential focus on the region around Y132)

3. LAT also contains a proline rich region (the senior author co-published on this). The proline rich region should also be covered in their comparison of different LAT orthologues.

6. PLOS authors have the option to publish the peer review history of their article (what does this mean?). If published, this will include your full peer review and any attached files.

Reviewer #1: No

---

## [Author Response · Author response to Decision Letter 0]

17 Feb 2023

Reviewer 1

LAT is a critical signaling adapter molecule in the T cell receptor signaling pathway that very likely plays a role in ligand discrimination. LAT orthologues have been identified in various species, including fish. Reports on LAT in bird species have been lacking.

In the submitted manuscript, the authors capitalize on their experience searching for and analyzing genes with high GC content. The authors describe the identification of chicken LAT and postulate – based on sequence homology to human LAT and chicken LAT expression – that chicken LAT full fills a similar function as mammalian in LAT in T cell signaling.

In sum, this is a clearly presented, brief definitive report on the identification of chicken LAT that is informative for the T cell signaling community.

We are thankful for the positive evaluation of the manuscript and useful comments.

The authors should enhance the manuscript in a couple of manners:

1. The presentation of the amino acid sequence alignment in figure 3 can be improved. It should contain the numbering of e.g. tyrosine residues that are mentioned in the text.

We added the numbering of the tyrosines.

2. The authors should also perform a comparison between chicken and fish LAT (with potential focus on the region around Y132)

We added the comparison of LAT from chicken and zebrafish (Fig. 3B).

3. LAT also contains a proline rich region (the senior author co-published on this). The proline rich region should also be covered in their comparison of different LAT orthologues.

We highlighted the proline-rich motif mediating LCK binding in the human sequence and also highlighted all prolines in the proline-rich sequence. It is interesting that the LCK-binding motif (PIPRSP) is missing in the chicken sequence, but there are still many prolines in the N-terminal part of the LAT intracellular sequence. We comment on this in the revised version of the manuscript.

---

## [Editor Report · Decision Letter 1]

10 Mar 2023

Identification of GC-rich LAT genes in birds

PONE-D-22-34260R1

Dear Dr. Stepanek,

We’re pleased to inform you that your manuscript has been judged scientifically suitable for publication and will be formally accepted for publication once it meets all outstanding technical requirements.

Kind regards,

Sebastian D. Fugmann, Ph.D.

Academic Editor

PLOS ONE
---

## [Editor Report · Acceptance letter]

29 Mar 2023

PONE-D-22-34260R1 

Identification of GC-rich LAT genes in birds 

Dear Dr. Stepanek:

I'm pleased to inform you that your manuscript has been deemed suitable for publication in PLOS ONE. Congratulations! Your manuscript is now with our production department. 

Kind regards, 

on behalf of

Dr. Sebastian D. Fugmann 

Academic Editor

PLOS ONE